# DMA Investigation of the Factors Influencing the Glass Transition in 3D Printed Specimens of Shape Memory Recycled PET

**DOI:** 10.3390/polym14112248

**Published:** 2022-05-31

**Authors:** Bogdan Pricop, Ștefan Dumitru Sava, Nicoleta-Monica Lohan, Leandru-Gheorghe Bujoreanu

**Affiliations:** 1Faculty of Materials Science, “Gheorghe Asachi” Technical University of Iași, Blvd. Dimitrie Mangeron 71A, 700050 Iasi, Romania; bogdan.pricop@academic.tuiasi.ro (B.P.); nicoleta-monica.lohan@academic.tuiasi.ro (N.-M.L.); 2Inotech Association, Plopii Fără Soț by Street 19A, 700281 Iasi, Romania; stefan@inotech.ngo

**Keywords:** recycled PET, 3D printing, shape memory effect, storage modulus, internal friction

## Abstract

Polyethylene terephthalate (PET) is used worldwide for packing, and for this reason, it is the main material in plastic waste. The paper uses granules of recycled PET (R-PET) as raw material for producing filaments for 3D printing, subsequently used for printing the test specimens in different ways: longitudinally and at angles between 10° and 40° in this direction. Both the filaments and the printed specimens experience thermally driven shape memory effect (SME) since they have been able to recover their straight shape during heating, after being bent to a certain angle, at room temperature (RT). SME could be reproduced three times, in the case of printed specimens, and was investigated by cinematographic analysis. Then, differential scanning calorimetry (DSC) was used, in R-PET granules, filaments and 3D printed specimens, to emphasize the existence of glass transition, which represents the governing mechanism of SME occurrence in thermoplastic polymers, as well as a recrystallization reaction. Subsequently, the paper investigated the 3D printed specimens by dynamic mechanical analysis (DMA) using a dual cantilever specimen holder. Temperature (DMA-TS) and isothermal scans (DMA-Izo) were performed, with the aim to discuss the variations of storage modulus and loss modulus with temperature and time, respectively.

## 1. Introduction

Polyethylene terephthalate (PET) is the most commonly used material for food packing and fluid transportation purposes, with a production estimated at 70 million tons in 2020 [1]. Therefore, the huge number of plastic bags and bottles produced has largely contributed to the accumulation of the present 1.6 million square kilometers of floating “Great Pacific Garbage Patch” [2]. Despite recycling in sustainable ways, at least a part of the giant amount of PET waste amassed since the first synthesis, which succeeded in the midst of the 1940s [3], has become one of the present time’s most critical challenges [4]. Therefore, a large number of papers dedicated to PET recycling were published, being focused on physical (thermal extrusion and pelletization) or chemical processes (aiming to decompose PET into depolymerized oligomers) [5]. Since PET degradation naturally occurs after prolonged exposures (to temperature, humidity, radiation or bacteria) [6], various approaches aimed to accelerate these processes, such as depolymerization at 110 °C for 5 h [7], melting and plasticization under shear and tension flows [8], or gamma rays’ exposure [9].

Among various modalities of PET recycling, the production of filament for 3D printing—also valorized in the case of other thermoplastics [10,11]—has gained increasing interest, especially after the start of its commercialization in 2015 [12]. Thus, comprehensive studies were performed to ascertain the suitability of recycled PET (R-PET) for the production of powders [13] and filaments for 3D printing [14]. It has been reported that printed parts with virgin and R-PET filaments had comparable mechanical properties [15], especially when they were obtained by fused filament fabrication [16].

Nevertheless, besides the possibility of being recyclable, PET has chemical resistance and excellent mechanical, processing and thermal properties [17] associated with its structure, comprising a crystalline and two amorphous fractions with different mobilities [18]. The investigations performed by dynamic mechanical analysis (DMA) emphasized a marked storage modulus decrease during heating above the thermal range for glass transition [17,19].

On the other hand, glass transition is a potential mechanism for the shape memory effect (SME) occurrence in thermoplastic polymers, which recover their permanent shape during heating after being deformed to a temporary one [20]. Moreover, storage modulus decreases by up to two orders of magnitude and is a criterion for the identification of thermal memory behavior [21]. Considering that PET has comparable specific work outputs with certain shape memory alloys [22], concrete crack closure systems have been successfully developed based on PET knotted fibers [23] of tendon outer sleeves [24].

This ability of 3D printed objects to alter their shape and function after being subjected to an external stimulus, such as temperature, has added an additional dimension to the additive manufacturing process, which became 4D printing [25], and evolved from a concept to a manufacturing paradigm [26].

Nevertheless, as far as the knowledge of the present authors, no reports have been published on SME occurrence in R-PET. Therefore, the present paper aims to emphasize SME presence, both in R-PET 3D filaments and printings, and to corroborate it with the thermodynamic behavior observed by calorimetry and dynamic mechanical analysis.

## 2. Materials and Methods

R-PET grains, originating from recipients of carbonated drinks bottles, were purchased from the GreenTech SA company, Buzău, Romania. As compared to virgin grains that are injected and blown into chilled molds, the grains used in the present study are heated, extruded into a filament, deposited and chilled. The grains were processed by the Inotech Association by means of the experimental line for 3D filament production, as illustrated in Figure 1.

After water removal in the grain dryer, the grains fell into the central unit, where they were heated up to 270 °C. Then, they were extruded through a dye into a 1.75 mm-diameter filament which passed through the hot water (80 °C) vat and the cold water vat, being drawn by the roll tractor. The rotation speed of the roll tractor was controlled by a laser system that read the filament’s diameter. Since the process was entirely continuous, before being wound into rolls, the filament was deviated by means of an accumulation line.

Using R-PET filament, parallelepipedal specimens were printed with the dimensions 1 × 4 × 50 mm. Five different angles between specimens’ longitudinal axis and filament deposition direction were used to print specimens at 0°, 10°, 20°, 30° and 40°, respectively. The images of the R-PET grains, the filament and the five types of specimens, with optical microscopy (OM) details, are illustrated in Figure 2.

The presence of the shape memory effect (SME) was investigated by means of cinematographic analysis [27]. Both filament fragments and printed specimens were bent at room temperature (RT) and subsequently heated with a hot-air gun while having a multimeter’s thermocouple fixed in the deformed area. Cooling was performed in free air. The variations of the specimen’s free end and temperature were recorded and analyzed frame-by-frame.

The heating induced changes were investigated by thermal analysis, comprising differential scanning calorimetry (DSC) and dynamic mechanical analysis (DMA).

The former was performed on a NETZSCH DSC 200 F3 Maia device (Netzsch, Selb, Germany), calibrated with Bi, In, Sn, Zn and Hg standards. Grains and fragments weighing less than 50 mg were cut from the filament and printed specimens and were heated under an Ar protective atmosphere at 10 °C/min.

The latter was achieved using a NETZSCH DMA 242 Artemis device equipped with a dual cantilever specimen holder. Printed specimens were dynamically bent under an Ar protective atmosphere during heating or isothermal testing. The former used a heating rate of 5 °C/min and 1 Hz frequency. The latter was performed at three temperatures, at three frequencies: 1, 10 and 100 Hz, and in each case, using a bending amplitude of 100 µm. Both DSC and DMA records have been evaluated by Proteus software, v.6.1, Selb, Germany.

## 3. Results and Discussion

### 3.1. Shape Memory Effect

The typical images of the free-recovery SME analysis, corresponding to the cold and hot shapes of the filament, are illustrated in Figure 3.

It is obvious that the R-PET filament experienced free-recovery SME during the first heating. After applying a supplementary RT bending, SME was not observed in the second cycle.

Similar images are shown in Figure 4 for the evolution of a printed specimen; however, in this case, the phenomenon was reproducible for three cycles. It is noticeable that the free end’s displacement remains more and more behind the variation of temperature.

In order to illustrate the free end’s displacement delay with regard to temperature, Figure 5 displays the shift of displacement-temperature diagrams after three cycles.

Figure 5 displays both experimental values, with filled symbols and fitted values, with solid lines, as a result of a Boltzmann fitting function. The general equation of the Boltzmann function has the following form:(1)$$y=A_{2}+(A_{1}-A_{2})/(1+e^{\frac{x-x_{0}}{dx}})$$

The values of the parameters of the Boltzmann function are listed in Table 1.

The standard errors, which do not exceed 1%, ascertain the validity of the mathematical model.

A possible cause for the SME’s disappearance in the filament after the first cycle might be the higher cooling rate of the filament during processing. The filament is extruded on its entire cross-section, drawn by the roll tractor and water-cooled while the specimens are printed layer-by-layer and air-cooled. Thus, a functionally-graded shrinkage along with the sample thickness was developed, which enhanced SME occurrence [28].

### 3.2. DSC Measurements

The representative DSC thermograms, recoded during heating, are summarized in Figure 6.

In most of the cases, the thermograms revealed three transformations: (i) glass transition, (ii) recrystallization and (iii) melting.

The grains did not reveal any recrystallization but seemed to experience two glass transitions (*T_g_*_1_ and *T_g_*_2_). The second transition might be an effect of ageing, underwent by original PET bottles before shredding. It has been reported that with the increasing ageing time [29], the degree of crystallinity increased and no recrystallization occurred [30]. It is noticeable, however, that both the recrystallization and melting became more intense in the printed specimens and even at the specimen printed at 40°, as compared to that printed at 0°. Figure 6b presents the thermograms of two specimens printed at 40°. The first was in an initial state and the second was subjected to three free-recovery SME cycles. It is obvious that the thermal range of glass transition moved to higher temperatures and the transition itself became less abrupt. Glass transition is directly related to the change of amorphous into crystalline phase and acts as a microstructural mechanism of SME [20]. During consecutive heating-cooling-bending cycles, when the temperature did not reach the recrystallization temperature, and cooling to RT was performed in air, one may suppose that the amount of amorphous phase decreased because low cooling rates do not enhance re-amorphization processes. Thus, it can be assumed that the glass transition underwent a degradation process caused by the decrease in an amorphous phase amount [18], which was reflected by the delay and final disappearance of the SME. This could sustain the displacement delay discussed in the previous section. A more comprehensive evaluation of the thermograms from Figure 6 is presented in Table 2.

The second glass transition was observed at grains, between 163.7 and 166.6 °C, and absorbed only 0.031 mJ/g °C. The grains, which were shredded from bottles cooled in water-chilled metallic molds, have a larger amount of amorphous matter, and this could be the reason why they absorb more energy during glass transition. It is obvious that the crystallization degree increased from the filament to the printed specimens and from 0 to 40° angle printing. In the last case, the crystallization degree decreased by 20%, from the first to the third free-recovery SME cycle.

### 3.3. DMA Measurements

The typical DMA diagrams recorded during heating are illustrated in Figure 7.

The sharp decrease in the storage modulus, seen in Figure 7a, is typical for the glass transition. Before the decrease, there is a local storage modulus increase, indicating the destabilization of the specimen. This increment has a minimum of 272 MPa for the specimen printed at 0° and a maximum of 839 MPa, for the specimen printed at 10°. The same two specimens had an extreme loss of modulus values of a maximum of 265 MPa at 0° and 417 MPa at 10°, as seen in Figure 7b. Figure 7c shows details of the storage modulus increase at the end of heating, which was previously reported in other articles, such as [8]. Considering the high mobility of the polymeric chains in the temperature range of 120–150 °C, the only possible explanation for a storage modulus increase can be associated with the recrystallization observed by DSC, as previously observed in a study on amorphous PET. In the same way, as in Figure 7c, the storage modulus began to rise at near 120 °C with a slower increasing rate and changed to a higher increasing rate near 130 °C [31].

The isothermal tests were performed at RT and at the temperature of storage modulus maximum. Figure 8 shows the storage modulus variation at RT for three 3D printed specimens.

As expected, the storage modulus values increased with test frequency and decreased with the augmentation of the deposition angle (raster) against the longitudinal direction of the specimen. The effect was explained, in the case of polylactic acid, by the change of the type and the number of deformed layers with increasing raster angles. Only individual layers were deformed at a 0° raster, while an increasing number of adjacent layers were deformed with an increasing raster angle [32]. According to the deposition-induced effect, the bonding area between the adjacent layers decreases with the increase of the raster angle [33].

In addition, at RT, the storage modulus remained almost constant in time. A simple calculation of the number of cycles at 100 Hz during 9 min gives 54,000 cycles.

A different situation can be observed for the values of storage modulus recorded isothermally, at the temperature corresponding to its minimum and maximum observed during heating, according to Figure 7. These variations are exemplified in Figure 9, for the specimen printed at 0°.

In this case, a strengthening effect can be observed for the storage modulus values recorded at 48 °C and 60 °C. At 48 °C, there was actually the start of the glass transition. It can be assumed, then, that the destabilization of the amorphous state just started and only a small part of the polymeric chains became crystalline. During 9 min, the storage modulus value increased by about 25%, probably because a part of the newly formed crystals became amorphous again. At 60 °C, there was actually a climax of glass transition. The amount of the crystalline part should be much larger than at 48 °C. Therefore, the material is softer than at RT when in the initial stage of the isothermal test. However, during the test, more and more became amorphous again and, for this reason, the storage modulus increases from approx. 1200 to 1600 MPa.

## 4. Conclusions

The following conclusions can be drawn:Free-recovery SME was emphasized, both in the case of the filaments produced from the R-PET pellets and in the case of the 3D printed parts obtained with these filaments.The printed parts experienced free-recovery SME for up to three consecutive cycles, during which a delay was noticed between the displacement and temperature variations, which were fitted with Boltzmann-type functions with standard errors below 1%. This delay was associated with glass transition degradation, probably caused by the decrease in the amorphous phase amount during free-air cooling.The DSC measurements emphasized a glass transition, which is the mechanism of SME and recrystallization which produced a storage modulus increase between 125 and 150 °C.After three SME cycles, degradations were observed on the DSC thermograms, both at the glass transitions and at recrystallization.The DMA measurements, performed with dual cantilever dynamic bending, emphasized the storage modulus increases during heating, before the glass transition thermal range.Increasing the angle between the specimen’s direction and layer deposition direction, from 0° to 30°, caused storage modulus decreases at RT due to the decrease in the bonding area between the adjacent layers, with an increased raster angle;Isothermal DMA measurements, performed at temperatures in the beginning and the climax of glass transition, emphasized the storage modulus increases in time by about 25%, which can be ascribed to the amorphization of a part of the newly formed crystallites.

## Figures and Tables

**Figure 1 polymers-14-02248-f001:**
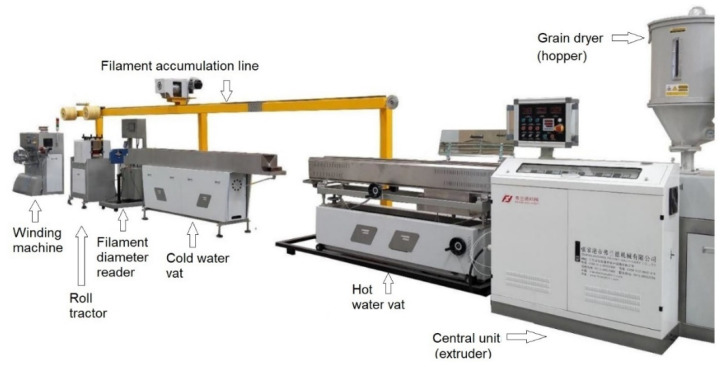
The experimental line for filament production from R-PET grains.

**Figure 2 polymers-14-02248-f002:**
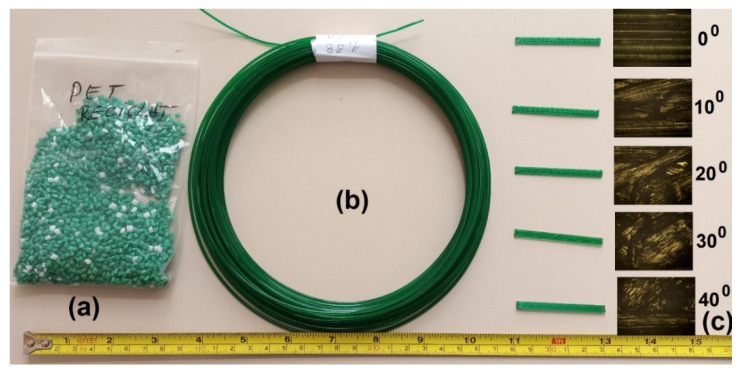
Images of raw materials and specimens: (**a**) R-PET grains; (**b**) 3D filament; (**c**) 3D printed specimens with corresponding OM details of the five different angles between their longitudinal axes and filament deposition direction.

**Figure 3 polymers-14-02248-f003:**
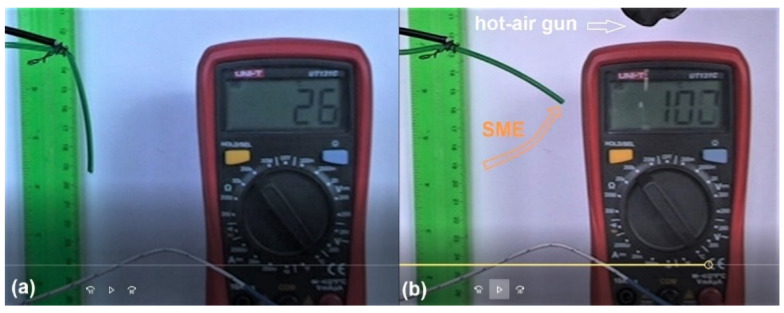
Typical images recorded during the occurrence of free-recovery SME at R-PET filament: (**a**) RT bent cold shape; (**b**) hot shape recovered during 23 s heating.

**Figure 4 polymers-14-02248-f004:**
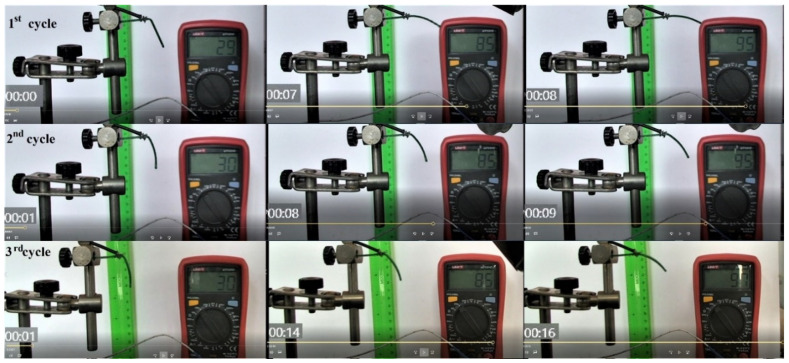
Typical images recorded during the occurrence of free-recovery SME at a 3D printed R-PET specimen during three consecutive heating-cooling-bending cycles. Time duration details (s).

**Figure 5 polymers-14-02248-f005:**
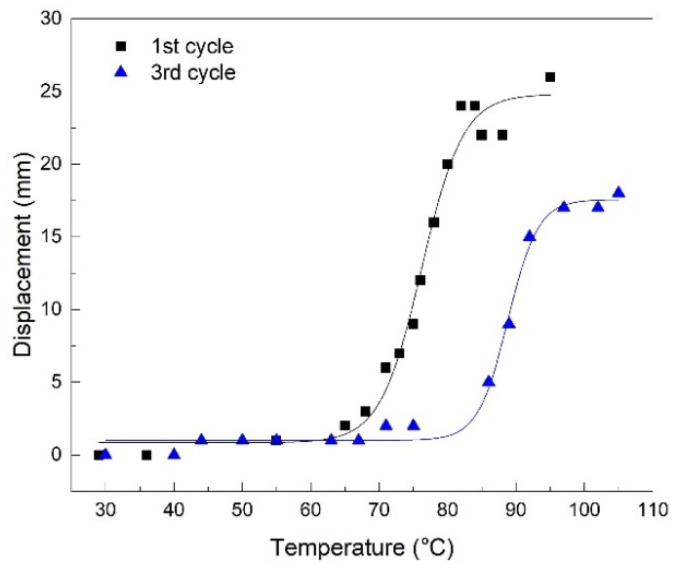
Free end’s displacement vs. temperature during free-recovery SME, according to Figure 4.

**Figure 6 polymers-14-02248-f006:**
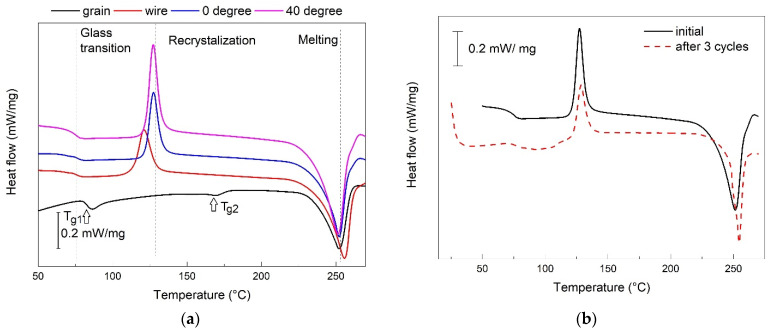
DSC thermograms recorded during heating: (**a**) R-PET grain, filament and printed specimens at 0° and 40°; (**b**) printed specimen at 40° in an initial state and after three free-recovery SME cycles.

**Figure 7 polymers-14-02248-f007:**
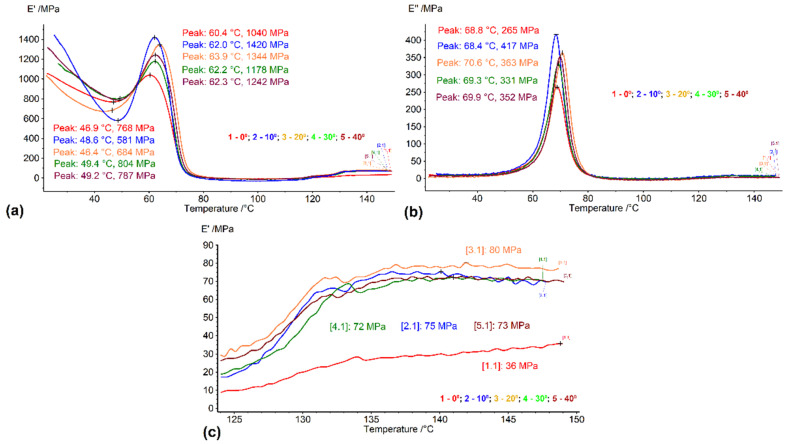
DMA diagrams recorded during the heating of 3D printed specimens: (**a**) variation in storage modulus; (**b**) variation in loss modulus; (**c**) details of the variation in storage modulus between 125 and 150 °C.

**Figure 8 polymers-14-02248-f008:**
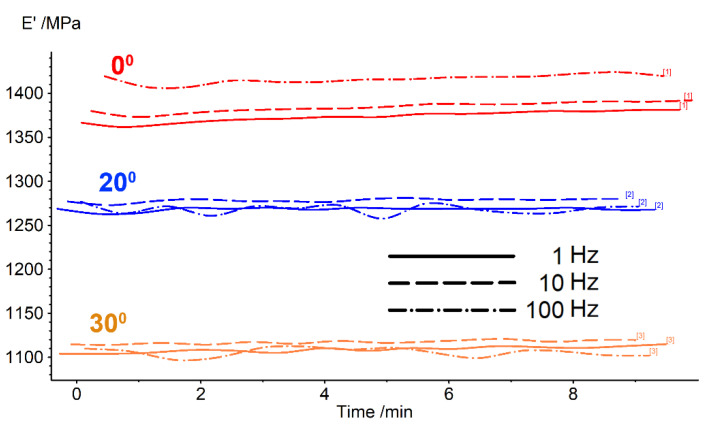
Isothermal DMA diagrams recorded at RT, with 3D printed specimens at 0°, 10° and 20° deposition angle.

**Figure 9 polymers-14-02248-f009:**
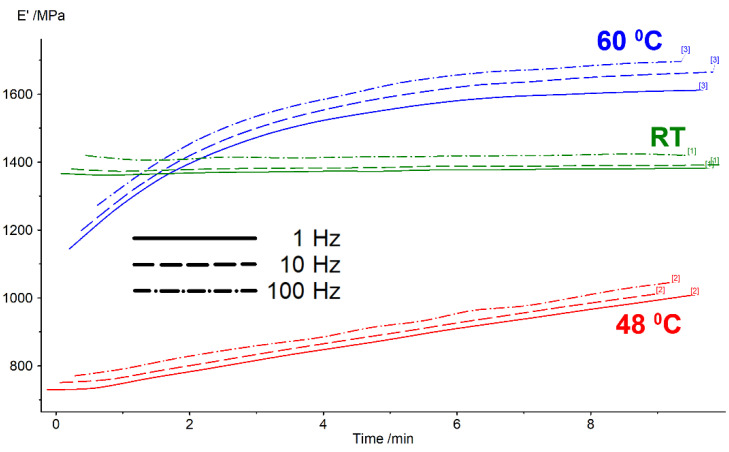
Isothermal DMA diagrams recorded at RT at 48 °C and at 60 °C with 3D printed specimens at 0°.

**Table 1 polymers-14-02248-t001:** Values of Boltzmann function parameters, according to the diagrams listed in Figure 5.

Parameter	1st Cycle		3rd Cycle	
	Value	Standard Error	Value	Standard Error
A_1_	0.84532	0.69398	1.00864	0.23239
A_2_	24.81075	0.93281	17.56472	0.43851
x_0_	76.20404	0.49517	88.82031	0.32189
dx	3.03517	0.45455	2.20105	0.31673

**Table 2 polymers-14-02248-t002:** Evaluation results of the thermograms from Figure 6.

Specimen	Glass Transition					Recrystallization
	Onset °C	Mid °C	Inflection °C	End °C	ΔC_p_ J/(g∙°C)	Δh kJ/kg
grain	80.4	82.6	83.1	84.4	0.279	-
filament	73.6	75.9	76.1	78.0	0.086	15.2
printed 0°	74.8	77.4	76.6	79.3	0.093	16.44
printed 40°	73.5	76.9	76.0	78.3	0.106	21.81
printed 40°, 3 cycles	74.4	76.8	77.2	79.3	0.101	17.44

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
