# Peer review of "DMA Investigation of the Factors Influencing the Glass Transition in 3D Printed Specimens of Shape Memory Recycled PET"

_polymers, 2022, doi:10.3390/polym14112248_

Round 1

Reviewer 1 Report

The article submitted by Pricop et al, entitled “DMA investigation of the factors influencing the glass transition in 3D printed specimens of shape memory recycled PET” (Polymers-172137), focuses on the topic of shape memory effect of 3D-printed PET by investigating the glass-transition. The results show that the storage modulus values varied with temperature and printing angle, the mechanism behind it were discussed. It is interesting. Overall, the article is well-prepared, and hence this reviewer suggests its publication. However, for the benefit of the reader, this reviewer would like to also point out a few issues. It would be better and further improve the impact of the paper if these issues are addressed by the authors before it is published.

  1. In this study, the recycling of PET should be not the main topic. In fact, in this reviewer’ opinion, why the author mentioned the recycling of PET, it is only because the used material is recycled PET. However, what is the difference of the shape memory effect between the recycled PET and new PET? What is the difference between the 3D Printing effect between both? These questions are not answered, and the authors also do not want to answer them at all. Hence, it is somewhat meaningless to overemphasize the recycled PET.

  1. Regarding to the used material, authors wrote “R-PET grains from recipients for carbonated drinks bottles were provided by the company GREENWEE International SA.” It is not clear. The details of the materials were not disclosed at all. What is the main difference between the recycled PET and the new one?

  1. In section of 3.1 shape memory effect, why is the shape memory effect of the RPET filament different from the printed specimen? The SME of the filament can be only found for one cycle, and the printed samples can have three cycles. The reason should be explained more.

  1. It is interesting to discuss the influence of the deposition angle. However, the reasons behind them are better to be explained more. Authors may refer to this article, Polymer Composites, 2020, 41, 60-72, and decide to discuss more about this concern.

  1. In total section of 3.3 DMA measurements, the mechanism of influence should be discussed more. For example, author mentioned, which can by associated with the recrystallization observed by DSC (L154), why? In last paragraph, L172-L181, some results were simply described, however, what can the reader obtain from these results? Further details are better to be discussed.

  1. In section of conclusion, some sentences are only the experimental results, not conclusion. For example, Increasing the angle between specimen’s direction and layer deposition direction, from 00 to 300, caused storage modulus decreases at RT.

In conclusion, some revisions are suggested before its publication.

Author Response

Thank you for your fair comments and suggestions. We did our best to respond to them, added 4 references, numbered [28] and [31]-[33] and reformulated the conclusions. Any line mentions refer to our revised submission.

Answers to the Comments and Suggestions for Authors, of Reviewer 1

  1. In this study, the recycling of PET should be not the main topic. In fact, in this reviewer’ opinion, why the author mentioned the recycling of PET, it is only because the used material is recycled PET. However, what is the difference of the shape memory effect between the recycled PET and new PET? What is the difference between the 3D Printing effect between both? These questions are not answered, and the authors also do not want to answer them at all. Hence, it is somewhat meaningless to overemphasize the recycled PET.

Answer

Indeed, PET recycling is not the topic. The main topic of the present paper is reporting for the first time, as far as our knowledge goes, the presence of SME at R-PET, which is a mean and not a purpose. Our co-author, Stefan Sava from Asociatia Inotech has been involved in developing innovative solutions for 3D printing with R-PET and other recycled polymers. It is this association that purchased the grains and produced the specimens and samples to be tested. We feel sorry if the Reviewer considered R-PET mentioning as an overemphatisation. It is only the raw material used in this study. But, considering that R-PET was mentioned 17 times in the manuscript, we removed this abbreviation from the text, at line 98and replaced it with Grains.

  1. Regarding to the used material, authors wrote “R-PET grains from recipients for carbonated drinks bottles were provided by the company GREENWEE International SA. It is not clear. The details of the materials were not disclosed at all. What is the main difference between the recycled PET and the new one?

Answer

Additional details were provided regarding the particularities of the R-PET grains in comparison to virgin ones, starting from line 69:R-PET grains, originating from recipients for carbonated drinks bottles, were purchased from the company GreenTech SA. As compared to virgin grains that are injected and blown into chilled molds, the grains used in present study are heated, extruded into a filament, deposited and chilled.

  1. In section of 3.1 shape memory effect, why is the shape memory effect of the R-PET filament different from the printed specimen? The SME of the filament can be only found for one cycle, and the printed samples can have three cycles. The reason should be explained more.

Answer

The following explanation, including reference [28] was added starting with line 132: “A possible cause for SME disappearance in the filament, after the first cycle, might be the higher cooling rate of the filament, during processing. The filament is extruded on its entire cross section, drawn by the roll tractor and water-cooled while the specimens are printed layer-by-layer and air-cooled. Thus, a functionally-graded shrinkage along with the sample thickness was developed which enhanced SME occurrence [28].

  1. It is interesting to discuss the influence of the deposition angle. However, the reasons behind them are better to be explained more. Authors may refer to this article, Polymer Composites, 2020, 41, 60-72, and decide to discuss more about this concern.

Answer

Supplementary explanations were added (together with references [31] and [32]), starting with Line 200: “As expected, storage modulus values increased with test frequency and decreased with the augmentation of the deposition angle (raster) against the longitudinal direction of the specimen. The effect was explained, in the case of polylactic acid, by the change of the type and number of deformed layers with increasing raster angle. Only individual layers were deformed at 0°raster while an increasing number of adjacent layers were deformed with increasing raster angle [31]. According to the deposition-induced effect, the bonding area between adjacent layers decreases with the increase of raster angle [32].”

  1. In total section of 3.3 DMA measurements, the mechanism of influence should be discussed more. For example, author mentioned, which can by associated with the recrystallization observed by DSC (L154), why? In last paragraph, L172-L181, some results were simply described, however, what can the reader obtain from these results? Further details are better to be discussed.

Answer

At Line 182 the following details were added, together with a new reference:Figure 7(c) shows a detail of storage modulus increase, at the end of heating, which was previously reported in other articles, such as [8]. Considering the high mobility of polymeric chains in the temperature range 120-1500C, the only possible explanation for storage modulus increase can by associated with the recrystallization observed by DSC, as previously observed in a study on amorphous PET. In the same way as in Figure 7(c), the storage modulus began to rise at near 1200C with a slower increasing rate and changed to a higher increasing rate at near 1300C [33].

  1. In section of conclusion, some sentences are only the experimental results, not conclusion. For example, Increasing the angle between specimen’s direction and layer deposition direction, from 0to 300, caused storage modulus decreases at RT.

Answer

The conclusion section was reformulated as follows:

“•          Free-recovery SME was emphasized both in the case of filaments produced from R-PET pellets and in the case of 3D printed parts obtained with these filaments;

  • The printed parts experienced free-recovery SME for up to three consecutive cycles, during which a delay was noticed between displacement and temperature variations, which were fitted with Bolzmann type functions with standard errors below 1 %. This delay was associated with glass transition degradation probably caused by the decrease of amorphous phase amount during free-air cooling;
  • DSC measurements emphasized a glass transition which is the mechanism of SME and a recrystallization which produced storage modulus increase between 125 and 1500C;
  • After three SME cycles, degradations were observed on DSC thermograms both at the glass transitions and at recrystallization;
  • DMA measurements, performed with dual cantilever dynamic bending, emphasized storage modulus increases, during heating, before glass transition thermal range;
  • Increasing the angle between specimen’s direction and layer deposition direction, from 00 to 300, caused storage modulus decreases at RT, due to the decrease of the bonding area between adjacent layers, with the increase of raster angle;
  • Isothermal DMA measurements, performed at temperatures of the beginning and the climax of glass transition, emphasized storage modulus increases in time, with about 25 %, which could be ascribed to the amorphization of a part of newly formed crystallites.”

Reviewer 2 Report

The manuscript, entitled “DMA investigation of the factors influencing the glass transition in 3D printed specimens of shape memory recycled PET”, by B. Pricop et al, investigated the thermal, mechanical, and shape memory behavior of 3D printed recycled PET materials. The authors employed a variety of characterization methods to evaluate PET performances to understand the structural change in recycled PETs. However, the authors need to address several issues to improve the quality of the manuscript before it is considered to be published in Polymers.

Major points:

  1. Experimental Details: Please include the details of experiments. For examples, how shape memory experiments were performed? The heated samples were naturally cooled to RT (or a different colling approach), and then got heated again? the time scale for performing one cycle of experiments (time is a critical factor)? DMA frequency range? DMA sample size? DMA strain? I would recommend including sub-sections in Materials and Method section and include more experiment details.
  2. Two glass transition temperatures: The authors discussed two glass transition temperatures, but the authors did not label these in Figure 6. Please label them. The authors provided explanation on this phenomenon but did not cite any literature in Discussion section to support these statements. Please cite papers properly to make the manuscript more scientifically sound.
  3. DMA details: The frequency range was 1-100 Hz, or 100-10000 Hz? It is not very clear to me. Please label Fig 9 more clearly. In the current version, the correlation between sample temp. and line color was not clear.
  4. Interpreting DSC Data: The authors should discuss DSC in a more quantitative manner. The authors need to integrate peaks to obtain exotherm values so the degree of crystallization could be compared quantitatively. Additionally, label Tg values in DSC curves.
  5. Figure 7: Please plot tan(delta) vs temperature as well.
  6. Degradation: the authors mentioned degradation after 3 cycles. Could the authors elaborate more? What have changed after three cycles? Physical or chemical degradation? Please cite papers properly and explain more.

Minor points:

Grammars and typos:

  • Line 62: “additional manufacturing” should be “additive manufacturing”
  • Please keep past and present tenses consistent. For example, line 139: “is” and “was” are confusing. Please proof-read.
  • Line 140: “larger temperatures” should be “higher temperatures”

Author Response

Thank you for your suggestions. We added any missing experimental details; two additional references, as [29] and [30] and Table 2. Figures 4. 8 and 9 were revised. We did our best as to respond to all your major comments, except for one, comment no.7. 

We could not plot tand vs temperature, due to a software error. Nevertheless, since tand = E"/E' and we plotted the variations of E" and E' we assume that all necessary details were provided to any potential reader.

Answers to the Comments and Suggestions for Authors, of Reviewer 2

Major points:

  1. Experimental Details:Please include the details of experiments. For examples, how shape memory experiments were performed? The heated samples were naturally cooled to RT (or a different colling approach), and then got heated again? the time scale for performing one cycle of experiments (time is a critical factor)? DMA frequency range? DMA sample size? DMA strain? I would recommend including sub-sections in Materials and Method section and include more experiment details.

Answer

The following details were added:

Line 92: “Cooling was performed in free air.” The rest of experimental details were provided in [27].

Line 112: “(b) hot shape recovered during 23-seconds heating”

Lines 120-121: Figure 4 was revised by adding magnified details of the time duration, in seconds. Additional text in caption: Line 123 “Time duration details (s)”

Line 102: “Printed specimens were dynamically bent under Ar protective atmosphere, during heating or isothermal testing. The former used a heating rate of 50C/min and 1 Hz frequency. The latter was performed at three temperatures, at three frequencies: 1, 10 and 100 Hz, in each case and used a bending amplitude was 100 µm. Both DSC and DMA records have been evaluated by Proteus software.” Specimen size was mentioned at line 81, of the original submission!

  1. Two glass transition temperatures: The authors discussed two glass transition temperatures, but the authors did not label these in Figure 6. Please label them. The authors provided explanation on this phenomenon but did not cite any literature in Discussion section to support these statements. Please cite papers properly to make the manuscript more scientifically sound.

Answer

Starting with line 147, the following explanations were added, together with two new references, [29] and [30]: “The grains did not reveal any recrystallization but seemed to experience two glass transitions (Tg1 and Tg2). The second transition might be an effect of ageing, underwent by original PET bottles before shredding. It has been reported that, with increasing ageing time [29] the degree on crystallinity increased and no recrystallization occurred [30].

At line 168, this sentence was added: “The 2nd glass transition was observed at grains, between 163.7 and 166.60C, and absorbed only 0.031m J/g0C.”

  1. DMA details: The frequency range was 1-100 Hz, or 100-10000 Hz? It is not very clear to me. Please label Fig 9 more clearly. In the current version, the correlation between sample temp. and line color was not clear.

Answer

The frequency ranges were clarified at Comment no. 1. Both Fig.8 and 9 were revised.

  1. Interpreting DSC Data:The authors should discuss DSC in a more quantitative manner. The authors need to integrate peaks to obtain exotherm values so the degree of crystallization could be compared quantitatively. Additionally, label Tg values in DSC curves.

Answer

The following sentence was added at Line 164: “A more comprehensive evaluation of thermograms from Figure 6 is presented in Table 2.” Then, Table 2 was added containing quantitative evaluation of DSC charts. At line 169: “The grains, which were shredded from bottles cooled in water-chilled metallic molds, have a larger amount of amorphous matter and this could be the reason why they absorb more energy during glass transition. It is obvious that the crystallization degree increased from filament to printed specimens and from 0 to 400-angle printing. In the last case, the crystallization degree decreased with 20 %, from the 1st to the 3rd free-recovery SME cycle.”

  1. Figure 7: Please plot tan(delta) vs temperature as well.

Answer

Due to the fact that, during glass transition, storage modulus decreased to zero, the PROTEUS software recognized this as an error and was unable to accurately plot tand variation with temperature. Sorry, we cannot submit those plots for publication. But, considering that tand is the ratio between loss modulus and storage modulus, any potential reader would get a comprehensive point of view about the phenomenon by exclusively monitoring the variation of the two moduli with temperature.

  1. Degradation: the authors mentioned degradation after 3 cycles. Could the authors elaborate more? What have changed after three cycles? Physical or chemical degradation? Please cite papers properly and explain more.

Answer

At line 156 the following explanations were added: Glass transition is directly related to the change of amorphous into crystalline phase and acts as a microstructural mechanism of SME [20]. During consecutive heating-cooling-bending cycles, when temperature did not reach recrystallization temperature, and cooling to RT was performed in air, one may suppose that the amount of amorphous phase decreased, because low cooling rates did not enhance re-amorphization processes. Thus, it can be assumed that glass transition underwent a degradation process caused by the decrease of amorphous phase amount [18] which was reflected by the delay and final disappearance of SME.

Minor points:

Grammars and typos:

  • Line 62: “additional manufacturing” should be “additive manufacturing”

Answer: Done

  • Please keep past and present tenses consistent. For example, line 139: “is” and “was” are confusing. Please proof-read.

Answer: Line 140: “larger temperatures” should be “higher temperatures”

Round 2

Reviewer 1 Report

I recommend its publication in present form. 

Reviewer 2 Report

I think the authors have attempted to answer all my questions properly. I really appreciated their patience and efforts.

I would recommend the publication of this manuscript at this point.